# Highly Sensitive and Rapid Quantitative Detection of *Plasmodium falciparum* Using an Image Cytometer

**DOI:** 10.3390/microorganisms8111769

**Published:** 2020-11-11

**Authors:** Muneaki Hashimoto, Kazumichi Yokota, Kazuaki Kajimoto, Musashi Matsumoto, Atsuro Tatsumi, Yoshihiro Nakajima, Toshihiro Mita, Noboru Minakawa, Hiroaki Oka, Masatoshi Kataoka

**Affiliations:** 1Health and Medical Research Institute, National Institute of Advanced Industrial Science and Technology (AIST), 2217-14, Hayashi-cho, Takamatsu, Kagawa 761-0301, Japan; kazumichi-yokota@aist.go.jp (K.Y.); k-kajimoto@aist.go.jp (K.K.); y-nakajima@aist.go.jp (Y.N.); m-kataoka@aist.go.jp (M.K.); 2Konica Minolta, 1 Sakura-mashi, Hino, Tokyo 191-8511, Japan; musashi.matsumoto@konicaminolta.com (M.M.); atsuro.tatsumi@konicaminolta.com (A.T.); hiroaki.oka@konicaminolta.com (H.O.); 3Department of Tropical Medicine and Parasitology, Juntendo University School of Medicine, 2-1-1 Hongo, Bunkyo-ku, Tokyo 113-8421, Japan; tmita@juntendo.ac.jp; 4Institute of Tropical Medicine, Nagasaki University, 1-12-4 Sakamoto, Nagasaki 852-8523, Japan; minakawa@nagasaki-u.ac.jp

**Keywords:** malaria, diagnosis, image cytometer, parasitemia

## Abstract

The gold standard for malaria diagnosis is microscopic examination of blood films by expert microscopists. It is important to detect submicroscopic and asymptomatic *Plasmodium* infections in people, therefore the development of highly sensitive devices for diagnosing malaria is required. In the present study, we investigated whether an imaging cytometer was useful for the highly sensitive quantitative detection of parasites. Whole blood samples were prepared from uninfected individuals spiked with *Plasmodium falciparum*-infected erythrocytes. Thereafter, erythrocytes were purified using a push column comprising of a syringe filter unit with SiO_2_-nanofiber filters. After adding the erythrocytes, stained with nuclear stain, to a six-well plate, quantitative detection of the parasites was performed using an image cytometer, CQ1. Imaging of 2.6 × 10^6^ erythrocytes was completed in 3 min, and the limit of detection indicated parasitemia of 0.00010% (≈5 parasites/μL of blood). In addition to rapid, highly sensitive, and quantitative detection, the ease of application and economic costs, image cytometry could be efficiently applied to diagnose submicroscopic parasites in infected people from endemic countries.

## 1. Introduction

According to the World Health Organization (WHO), in 2018, *Plasmodium falciparum* accounted for 99.7% of estimated malaria cases in Africa, the most endemic area of malaria. If left untreated, within 24 h malaria can progress to severe illness, often causing mortality [1]. Examination of Giemsa-stained blood films by expert microscopists (Giemsa microscopy) is the gold standard for diagnosing malaria in symptomatic individuals. At present, because endemic areas lack fully trained microscopists, rapid diagnosis tests (RDTs) have gained popularity. RDTs require minimal training and equipment (i.e., optical microscope with 100× objective lens); however, they are qualitative and detect residual malarial antigens leading to misdiagnoses [2,3].

Recent reports have revealed that numerous parasite carriers maintain low levels of parasitemia, thereby increasing the need for highly sensitive diagnostic devices [4,5]. Nucleic acid amplification tests (NAATs) such as PCR or loop-mediated isothermal amplification (LAMP) are performed for detecting submicroscopic and asymptomatic carriers [6]. Because quantitative PCR (qPCR) is highly sensitive, it is often performed to detect parasites in the submicroscopic carriers [7]; however, the cost of qPCR is extremely high as it requires an expensive thermal cycling, high-speed centrifuge for the purification of DNA from blood samples, and a clean bench to avoid contamination. Furthermore, fully trained laboratory technicians are required to perform such molecular biology techniques. Therefore, the application of qPCR is limited to laboratories in developed countries [6].

Previously, we developed diagnostic devices that quantitatively detect parasites marked with nuclear stain following erythrocyte purification with SiO_2_-nanofiber (NF) filters and forming monolayers of numerous erythrocytes on plastic substrates [8,9,10]. Although the number of erythrocytes forming a monolayer on cell microarray chips was unstable, stable erythrocyte monolayer formation could be achieved by using a fluorescent Blu-ray optical system, and the utility of these devices in Africa has been reported [11,12]. On average, 0.87 × 10^6^ erythrocytes formed a monolayer on a compact disc-like cassette of the device. Approximately 40 min of imaging was required to evaluate parasitemia.

In the present study, we report a more sensitive and rapid diagnostic test. Whole blood from healthy donors spiked with in vitro cultured parasite-infected erythrocytes was prepared, and parasitemia was estimated using an image cytometer, indicating that 2.5 × 10^6^ erythrocytes could be imaged for quantitative detection of the infected parasites in 3 min. Image cytometry may contribute to a highly sensitive and rapid diagnosis of submicroscopic infected carriers. The development of point-of-care testing (POCT) devices for many diseases is required [13]. Malaria diagnosis using image cytometry has the potential to replace the present qualitative POCT using RDT, hopefully leading to the elimination of malaria.

## 2. Materials and Methods

### 2.1. Parasite Culture

*P. falciparum* (3D7 strain) was cultured as previously described [9] and was synchronized at the ring stage, as previously described [14]. Parasitemia was estimated by microscopic analysis of Giemsa-stained thin blood films. More than 40,000 erythrocytes (red blood cells, RBCs) were enumerated to estimate parasitemia [% parasitemia = (parasitized RBCs/total RBCs) × 100].

### 2.2. Preparation of Push Columns

Empty syringe filter units (13 mm) were purchased from Pepaless Co., Ltd. (Hyogo, Japan). SiO_2_-NF filters (250 µm) (Blood separation filter LF1, GE Healthcare, Chicago, IL, USA) were punched at 13 mm diameters. Two filter sheets were set in the syringe filter unit (TERUMO Corporation, Tokyo, Japan) (Figure 1). The filter unit was set to 10 mL to prepare a push column.

### 2.3. Erythrocytes, Leucocytes, and Platelet Enumeration

Fresh whole blood samples drawn by finger pricking from healthy Japanese volunteers who had never been infected with malaria parasites were collected in BD Microcontainer blood collection tubes (K2EDTA, Lavender, Becton, Dickinson Co., Franklin Lakes, NJ, USA). Each whole blood sample (200 µL) was diluted 25-fold with phosphate-buffered saline (PBS) and then filtered using a push column. To enumerate the erythrocytes, the filtered blood sample was diluted 100-fold with PBS, and 10 µL of the aliquot was applied to a photon slide (Logos Biosystems, Inc., Gyeonggi-do, Korea). Erythrocytes were enumerated with a LUNA-FL cell counter (Logos Biosystems, Inc.) using the “bright field counting mode”, as described in the manual. Leucocytes in the filtered blood samples were stained with an AO/PI Cell Viability Kit (Logos Biosystems, Inc.), and 10 µL aliquots were applied to photon slides. Leucocytes were enumerated with a LUNA-FL counter using the “fluorescence cell counting mode”. The exposure times for the green fluorescence and red fluorescence were set to levels 5 and 1, respectively. To detect leucocytes at their respective exposure times, the threshold for both green fluorescence and red fluorescence in the protocol menu was set to level 5, and size gating was 5–30 µm. Platelets were enumerated using a flow cytometer (BD FACS Calibur, Becton, Dickinson Co.) with a fluorescein isothiocyanate (FITC)-labeled anti-human CD61 monoclonal antibody (Beckman Coulter, Inc., Brea, CA, USA). The percentage of platelets in the blood was estimated by counting 100,000 blood cells.

### 2.4. Detection of Parasite-Infected Erythrocytes Using CQ1

To analyze the parasite-infected erythrocytes using a confocal image cytometer CQ1 (Yokogawa Electric Corporation, Tokyo, Japan), an appropriate volume of purified parasite-infected erythrocyte suspension was added to whole blood at 6 × 10^6^ erythrocytes/mL. Erythrocytes were purified from 10 mL of each blood sample using a push column (Figure 2A). Filtered samples were diluted to 2.5 × 10^6^ erythrocytes/mL with PBS, and 10 μL of Cellstain AO solution (1 mg/mL, Dojindo Laboratories, Kumamoto, Japan) was added to stain the parasites. Then, 2 mL of this diluted sample was placed on 6-well plates (No. 140675, Thermo Scientific Nunc, Waltham, MA, USA) (Figure 2B). The plates were allowed to stand for 10 min to sediment the erythrocytes and then placed in the CQ1 cytometer (Figure 2C). Imaging was performed with a 4× objective lens and a fluorescence filter (excitation at 488 nm, emission at 525/50 nm), imaging of the bright field (excitation laser power of 80%, exposure time of 50 ms), fluorescence (excitation laser power of 100%, exposure time of 50 ms), and imaging of the 49 frames (7 × 7 frames) at the center of the well.

The parasites were quantitatively detected using CellPathfinder software (Yokogawa Electric Corporation) (Figure 2D). The algorithm for the detection of fluorescent images was set as follows: Division Mode “none”, minimum gray offset [gray level] “80”, range of area “10 to 100 μm^2^”. Because 52% of the whole bottom face in the well was imaged by the aforementioned settings, 2.6 × 10^6^ erythrocytes were analyzed. Thereafter, parasitemia was calculated using the following equation: parasitemia = [number of detected fluorescent signals]/[2.6 × 10^6^] × 100 (%).

### 2.5. Statistical Analysis

Statistical analysis was performed with SigmaPlot Version 13 (Systat Software, Inc., San Jose, CA, USA). The Student’s *t*-test was used to compare the mean number of blood cells and platelets before and after the filtration of whole blood samples. Statistical significance was set at *p* < 0.05. To determine the limit of detection (LOD), uninfected blood samples after filtration were analyzed ten times using CQ1, and the standard deviation (SD) of the background was calculated. Next, filtrated blood samples spiked with parasite-infected erythrocytes whose parasitemia was determined by Giemsa microscopy were analyzed five times using CQ1. Linear regression analysis was performed to calculate the coefficient of determination (R^2^) and the slope value of the approximation straight line (S). The LOD was determined to be 3SD/S.

### 2.6. Ethics Statement

Ethics approval was obtained from the National Institute of Advanced Industrial Science and Technology, Japan ethics committee (No. 2018-0204) on 4 December 2019.

## 3. Results and Discussion

### 3.1. Development of Push Columns for the Purification of Erythrocytes

Malaria can be diagnosed using devices that can detect unicellular parasites in erythrocytes, and the purification of erythrocytes from whole blood is crucial for good performance of such devices. In particular, the contamination of samples by leucocytes and platelets that are comprised of nucleic acids may increase the background signals. Recently, purification of erythrocytes from whole blood samples, without parasitemia alteration, was reported to be possible by using spin columns with SiO_2_-NF filters [10]; however, a centrifuge is required for the spin column, and its usage is restricted in several endemic areas where the power supply is insufficient. In order to purify erythrocytes in endemic areas, a push column was developed.

Diluted blood samples were applied to the push columns, and the filtered sample resulted in purified erythrocytes. The erythrocyte count enumerated before filtration was 4.9 ± 0.62 × 10^6^ erythrocytes/µL, and was 3.6 ± 0.53 × 10^6^ erythrocytes/µL after filtration (Figure 3A). Upon filtration, 25.6% of erythrocytes were removed on average. Moreover, the leukocyte count before filtration was 8.2 ± 1.2 × 10^3^ cells/µL, and no leucocytes were detected after filtration using this assay method (Figure 3B). Furthermore, the platelet count in the samples was also enumerated using a flow cytometer with an anti-CD61 antibody (Figure 3C). The platelet count before filtration was 2.3 ± 0.13 × 10^5^ platelets/µL, and was 0.013 ± 0.0034 × 10^5^ platelets/µL after filtration. After filtration, almost all leucocytes were removed, and 99.4% of the platelets were removed on average.

Filtration was performed at various speeds (i.e., various pressures). The purity of erythrocytes remained unaltered, and no hemolyzation was detected. Therefore, stable purification of erythrocytes using a push column may be possible without expert assistance. Collectively, it was possible to remove leukocytes and platelets from whole blood easily without a power supply by using a push column with SiO_2_-NF filters.

### 3.2. Detection of Parasites Using CQ1

Typical images of whole blood samples diluted at 5 × 10^6^ erythrocytes/well with PBS before and after filtration with the push columns are illustrated in Figure 4A,B, respectively. Typical images of whole blood spiked with a parasite culture diluted at 5 × 10^6^ erythrocytes/well with PBS before and after filtration are illustrated in Figure 4C,D, respectively. The bright-field (upper panels) and fluorescence images stained with AO (lower panels) of each sample are depicted.

For precise quantitative parasite detection using CQ1, erythrocytes have to form a monolayer, since only parasites infecting erythrocytes attached to the bottom of the well can be detected by this imaging technique. The bright-field images indicated that erythrocytes were not densely distributed and they did not form multilayers. Therefore, the area of the well was shown to be wide enough for 5 × 10^6^ erythrocytes. While leukocytes were detected in whole blood samples (Figure 4A), they were removed by filtration (Figure 4B). Because the size of the parasites’ nuclei (Figure 4E, arrows) were remarkably smaller than those of the leukocytes (Figure 4E, arrowhead), these were distinguishable from each other, and the parasite-infected and uninfected erythrocytes could be analyzed using CQ1.

### 3.3. Quantitively and LOD of CQ1 for Parasitemia Calculation

To investigate the quantitative data and the LOD of CQ1 for estimating parasitemia, whole blood samples spiked with parasite cultures with different percentages of parasitemia were prepared. The final parasitemia of the samples was 0.009%, 0.0045%, 0.0009%, and 0%. Each sample was filtered with an SiO_2_-NF spin column, followed by staining with AO, and parasitemia was estimated using CQ1 as described.

Each of the samples with 0% parasitemia was analyzed ten times, whereas the other samples were analyzed five times each. Parasitemia was estimated by Giemsa microscopy and CQ1, and both were compared (Figure 5). Although Giemsa microscopy with thin blood films is the gold standard method for the determination of parasitemia [2], standardized protocols for the determination of parasitemia in submicroscopic carriers are currently lacking [6]. Blood samples with very low parasitemia were prepared by diluting blood samples with elevated parasitemia, therefore Giemsa microscopy was the most appropriate method of reference for comparison. The coefficient of determination (R^2^) was 0.89, and the background by LUNA-FL was 0.00010 ± 0.000044%. LOD was determined as 3SD/S (SD = standard deviation of the background; S = 1.28, the slope value of the approximation straight line), and was found to be 0.000010% (≈5 parasites/μL of blood), which was comparable to that of qualitative nested PCR or LAMP [6]. We have reported the LOD of nested PCR under the same laboratory conditions [14]. When filter paper (Whatman 3MM Chr paper) was used for the preparation of dried blood spots, the LOD was determined to be 5.5 parasites/μL of blood.

A malaria diagnosis device, Parasight, has been developed and sold by Sight Diagnostics Ltd. (Jerusalem, Israel) [15]. Parasight works with diluted blood samples that have to be applied to exclusive loading cartridges, followed by monolayer formation of fluorescent-labeled blood cells. After imaging, parasitemia is estimated using the software developed by Sight Diagnostics Ltd. This diagnostic device offers rapid imaging at the same level as that of CQ1 and has a low detection limit (20 parasites/μL). Therefore, Parasight might be an ideal diagnostic device for accurate diagnoses in POCT in hospitals in Africa. However, it has not been introduced extensively across Africa. Parasight has three light-emitting diode (LED) light sources, and the easy-to-use but expensive loading cartridges have to be purchased regularly. Therefore, not only the initial cost but also the running costs of Parasight may prove to be too expensive for effective malaria diagnosis in Africa.

In Africa, *P. falciparum* is highly abundant compared to other *Plasmodium* species. In the present study, we demonstrated that highly sensitive and quantitative detection of *P. falciparum* parasitizing blood could be performed using SiO_2_-NF filters and an imaging cytometer. Although CQ1 is too expensive to use in malarial endemic areas, it is very effective as a parasite-detecting device. For example, although confocal laser scanning can be performed using CQ1, the imaging in this study was performed with a 4× objective lens for normal fluorescent imaging. Moreover, only one LED light source was used in this study. Collectively, quantitative detection of the parasites could be performed using simplified image cytometers, presumably reducing the initial cost. Furthermore, the method described in this study did not require an exclusive cartridge, and common culture plates were used for parasite detection. It is noteworthy that these plates are inexpensive (USD 2/6 samples), leading to a low running cost.

As a drawback of the analysis using an image cytometer, erythrocytes have to form a monolayer and the integrity of erythrocytes is crucial. Therefore, the samples after blood collection must be analyzed in field settings. On the other hand, blood films for Giemsa microscopy and filter papers containing blood samples for NAATs can be preserved, and they can be taken from the field and analyzed in fully equipped laboratories in developed countries. If appropriate storage procedures for erythrocytes are developed, the image cytometer technique may be significantly improved. This study is very important for mass screening in epidemiological surveys, because a large number of blood samples may be investigated simultaneously.

Babesiosis is a malaria-like parasitic disease caused by infection with eukaryotic parasites from the order Piroplasmida and the phylum Apicomplexa; typically, *Babesia* or *Theileria* [16]. Leishmaniasis, human African trypanosomiasis, and Chagas disease are caused by the taxonomically related kinetoplastid protozoan parasites, *Leishmania* spp., *Trypanosoma brucei*, and *T. cruzi*, respectively [17]. Because they parasitize blood cells or the bloodstream, image cytometry could also be useful for their quantitative detection.

Importantly, our experimental procedure, including erythrocyte purification by push columns, is much easier than methods requiring qPCR, trained experts, and molecular biology laboratories. This study is a proof of concept for the highly sensitive detection of *P. falciparum* using image cytometers. When devices applied to this methodological principle are developed, the detection of submicroscopic parasites in infected patients in Africa would be possible, thereby contributing to blocking the transmission of the parasites.

## Figures and Tables

**Figure 1 microorganisms-08-01769-f001:**
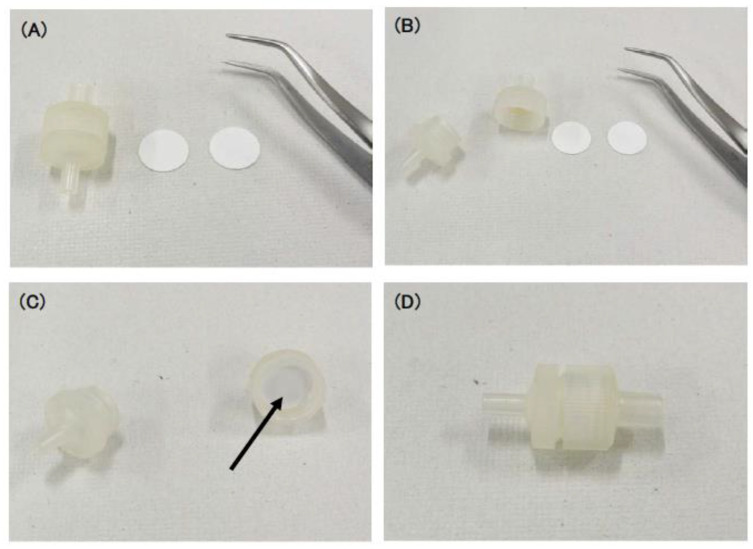
Setting SiO_2_-nanofiber (NF) filters in a syringe filter unit. (**A**) A syringe filter unit (left), two sheets of punched SiO_2_-NF filters (middle), and a tweezer (right). (**B**) Opened syringe filter unit. (**C**) Setting two sheets of the SiO_2_-NF filters (arrow) with the tweezer. (**D**) Closed syringe filter unit.

**Figure 2 microorganisms-08-01769-f002:**
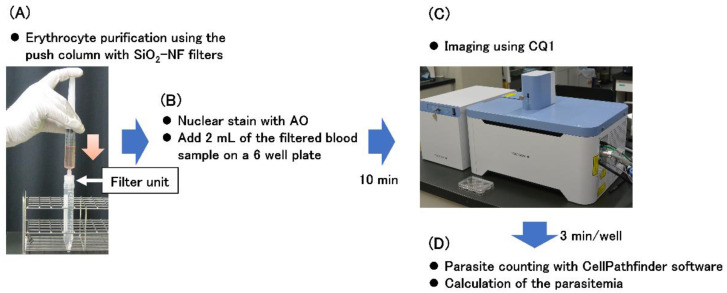
Protocol for highly sensitive quantitative detection of *Plasmodium falciparum* in blood using CQ1. Erythrocyte purification from diluted blood sample using the push column with SiO_2_-NF filters (**A**) followed by staining nuclei of the parasites was performed, and then 2 mL of the filtered blood sample was added on a 6 well plate (**B**). After standing for 10 min, imaging was performed using CQ1 (**C**). Parasitemia was calculated with CellPathfinder software (**D**). Details are described in the Materials and Methods section. Six blood samples were simultaneously analyzed. Parasitemia was estimated within 20 min after blood sampling.

**Figure 3 microorganisms-08-01769-f003:**
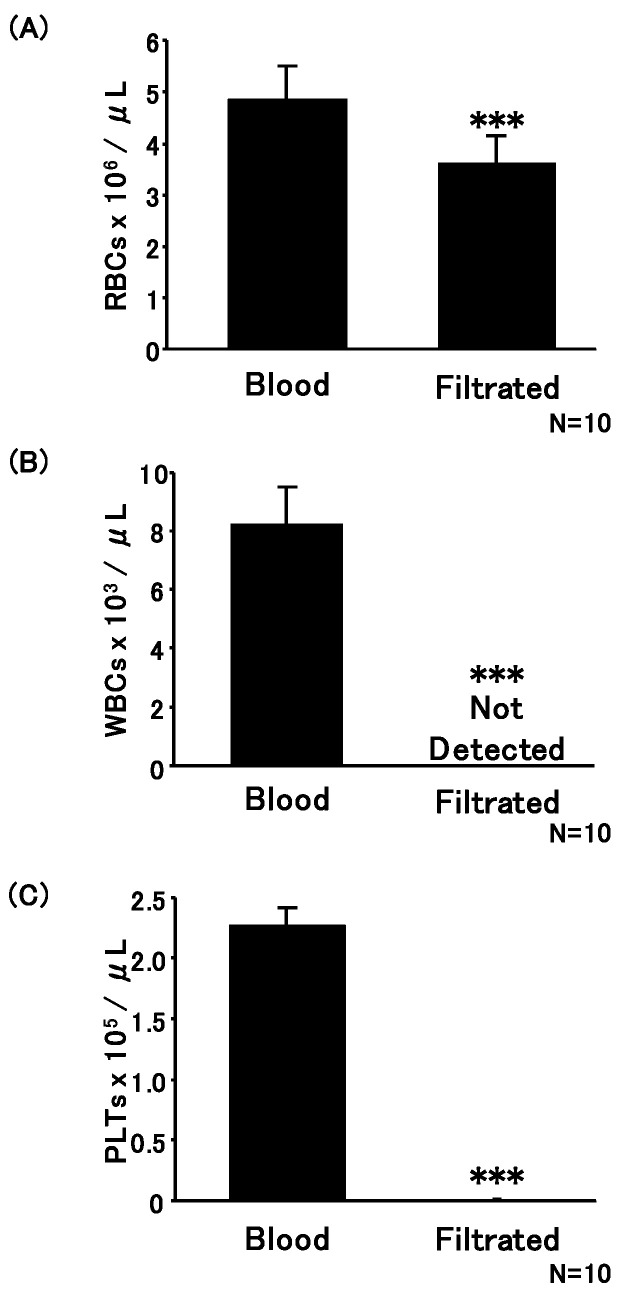
Evaluation of push column performance in the purification of erythrocytes from whole blood. The number of erythrocytes (red blood cells, **A**), leucocytes (white blood cells, **B**), and platelets (PLTs, **C**) was analyzed before (left bars) and after (right bars) filtration of whole blood. *** *p* < 0.001 (Student’s *t*-test, *n* = 10).

**Figure 4 microorganisms-08-01769-f004:**
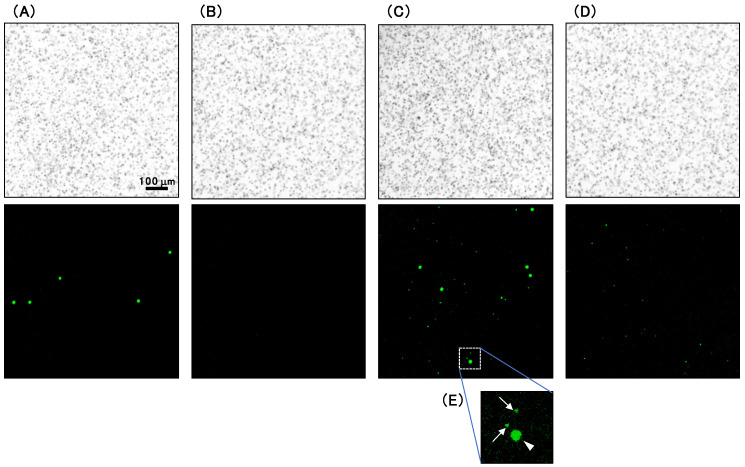
Bright-field and fluorescent images captured using CQ1. Whole blood (**A**), whole blood after filtration (**B**), whole blood spiked with parasite-infected erythrocytes (**C**), and whole blood spiked with parasite-infected erythrocytes after filtration (**D**) was analyzed via CQ1. Each sample was stained with Cellstain AO solution. Typical bright-field (upper panels) and fluorescence images (lower panels) are depicted. (**E**) An enlarged image of (**C**) is presented. A leucocyte is indicated by an arrowhead, and parasites are indicated by arrows.

**Figure 5 microorganisms-08-01769-f005:**
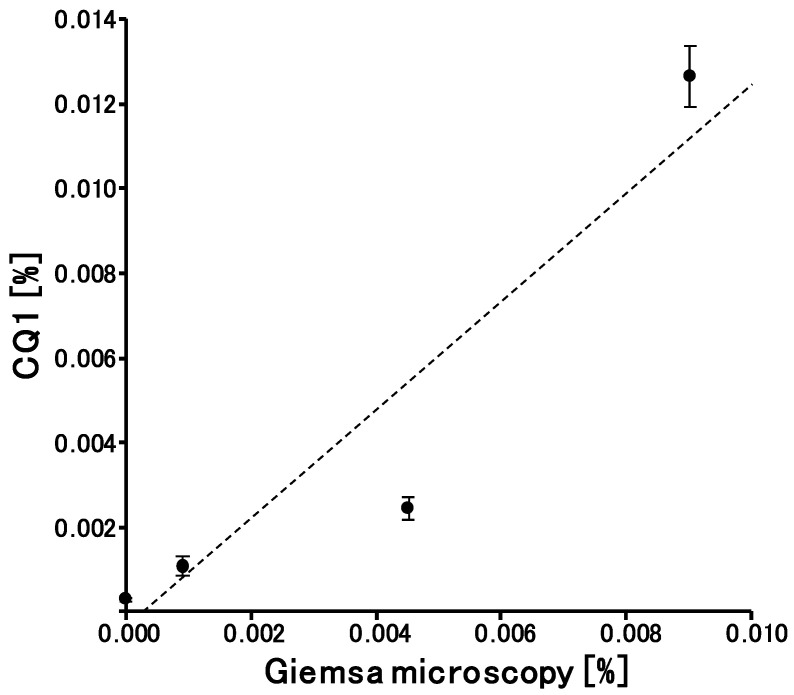
Comparative analysis of the parasitemia estimated with CQ1 and Giemsa microscopy. Linear regression analysis was used. Data are expressed as the mean ± SD for five different experiments.

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
