# Peer review of "Highly Sensitive and Rapid Quantitative Detection of Plasmodium falciparum Using an Image Cytometer"

_microorganisms, 2020, doi:10.3390/microorganisms8111769_

Round 1

Reviewer 1 Report

Summary of Study. This study investigated whether an imaging cytometer was useful for a high sensitive quantitative detection of Plasmodium falciparum malaria parasites. The authors of this study spiked blood samples from naïve, malaria free individuals with lab strain of P. falciparum 3D7 and used an image cytometer to provide quantification of the parasitemia. They report that it took 3 minutes for the imaging. They also report on the effectiveness of a push filtration system to purify erythrocytes by removing leucocytes and platelets.

Overall.

The manuscript reports simple experiment to demonstrate a proof of concept that a flow cytometer could be used for quantification of sub-microscopic malaria parasites.  It is a straightforward manuscript, that is easy to read although it had a few corrections and mistakes that need to be rectified.

Specific comments.

  • Please outline the specific statistical analysis that were done. Please include these under the 2.5 – Statistical analysis section.
  • If the intent was to show proof of concept that this new method is better than a gold standard, there should have been a head to head comparison between the new method and a standard methodology for sub-microscopic parasite detection such as PCR. I don’t want to take away from the excellent work done by the authors and the possibilities of this new method in parasite detection, but it has to be compared with widely used technologies. This work was done in a lab setting, and so it could have been easily been compared with standard methods.
  • Also, the requirement that the erythrocytes form a monolayer is fundamental to the CQ1 technique to accurately quantify malaria parasites, but it’s not clearly pointed out as a drawback compared to the other techniques. It appears this study has attempted to highlight the positives of the CQ1 and diminish the negatives. It would be better to set clear parameters to compare this method with the other well-established methods.

Author Response

Please outline the specific statistical analysis that was performed. Please include these under the 2.5 – Statistical analysis section.

Response

    An outline of the statistical analysis is described in the revised Statistical analysis section (lines 128-135).

Comment 2

If the intent was to show proof of concept that this new method is better than a gold standard, there should have been a head-to-head comparison between the new method and a standard methodology for sub-microscopic parasite detection, such as PCR. I do not want to take away from the excellent work done by the authors and the possibilities of this new method in parasite detection, but it has to be compared with widely used technologies. This work was done in a lab setting, and so it could have been easily compared with standard methods.

Response

    Thank you very much for appreciating our study. As suggested, in this study, parasitemia was determined using the Giemsa microscopy in the context of thin blood smears. The most precise parasitemia can be calculated using this method. To determine the LOD of the CQ1 technique, blood samples with very low parasitemia were prepared via the diluting blood samples with significantly high parasitemia. Therefore, we think that the parasitemia was accurate even if the value was very low, and that Giemsa microscopy was appropriate in this study. As you are concerned because diluted parasite-infected blood samples could not be prepared in field settings (i.e., real blood samples from infected people, not in laboratories), NAATs such as PCR should be performed as the reference methods.

On the other hand, we recently reported the LOD of nested PCR in the same laboratory environment (Hashimoto et al., Parasitol Int, 2019). The LOD was determined at 5.5 parasites/μL when filter papers (Whatman 3MM Chr papers) were used for the preparation of dried blood spots, indicating that the LOD of the CQ1 technique and that of nested PCR were similar. These important points have been added to the revised manuscript (lines 202-204).   

Comment 3

The requirement that the erythrocytes form a monolayer is fundamental to the CQ1 technique to accurately quantify malaria parasites, but it is not clearly pointed out as a drawback compared to the other techniques. It appears that this study has attempted to highlight the positives of CQ1 and diminish the negatives. It would be better to set clear parameters to compare this method with other well-established methods.

Response

    As suggested, the drawback of the image cytometer technique has not been fully described in the previous manuscript. Blood films for Giemsa microscopy and filter papers containing blood samples for NAATs can be preserved, and they can be taken from the field and can be analyzed at fully equipped laboratories in developed countries. On the other hand, erythrocytes have to form a monolayer and the integrity of erythrocytes is crucial for the analysis using the image cytometer. Then, the samples after blood collection must be analyzed in the field settings. If appropriate storage procedures for erythrocytes are developed, such issues may be no longer be a problem; however, that is not possible at the present. This study may be very important for mass screening in epidemiological surveys, since a large number of blood samples can be investigated simultaneously. The above points are described in the revised manuscript (lines 231-238).       

We would like to thank you and the referees for the helpful comments.

Reviewer 2 Report

The authors have provided an excellent proof of the concept of diagnosing Plasmodium falciparum using an image cytometer. The method is fast as it takes just 3 min and the LOD of 5 parasites per uL of blood samples is great. The method would be of great relevance to diagnostic settings due to its simplicity and good bioanalytical performance.

I have a few minor suggestions to improve the draft.

  1. The authors should write a few lines on how their rapid method could lead to prospective POCT devices and platforms. This could be added to the introduction. A useful ref here would be Trends in Biotechnology 33(11), 692-705, 2015.
  2. Several smartphone-based flow cytometry and imaging devices have been developed by different groups. Would these be of relevance and could be employed with this method? If yes, please add a few lines on this in the discussion section. A useful ref here would be Analytical and bioanalytical chemistry, 406(14), 3263-3277, 2014. Prof Ozcan group at UCLA has developed several such devices.
  3. Perhaps, the method would be of relevance for the detection of other parasites too based on the same approach. The authors can provide some future direction and guide the researchers towards it.
  4. The clinical evaluation trial of the method with an established predicate method used by healthcare would add much more value to the study. This could be a subject for future investigation by the authors. If the authors want, they can specify it in the text.

Author Response

Comment 1

The authors should write a few lines on how their rapid method could lead to prospective POCT devices and platforms. This could have been added to the introduction. A useful reference here would be Trends in Biotechnology 33(11), 692-705, 2015.

Response

    Thank you very much for appreciating our study and for introducing the very important review paper. As per your suggestion, we have described how image cytometry is considered a prospective POCT device and platform in the Introduction section of the revised manuscript (lines 65-68).  

Comment 2

    Several smartphone-based flow cytometry and imaging devices have been developed by different groups. Would these be of relevance and could be employed with this method? If yes, please add a few lines on this in the Discussion section. A useful reference here would be Analytical and Bioanalytical Chemistry, 406(14), 3263-3277, 2014. The Prof Ozcan group at UCLA has developed several such devices.

Response

Smartphone-based diagnosis devices for malaria are under development (for example, Rosado et al., Mobile-Based Analysis of Malaria-Infected Thin Blood Smears: Automated Species and Life Cycle Stage Determination. Sensors (Basel) 2017 Sep 21;17(10):2167. doi: 10.3390/s17102167.). These devices will help the diagnosis via Giemsa microscopy. However, skilled technicians are required for the preparation of blood films for the method, and the LOD of smartphone-based devices is not high; these devices have been developed for symptomatic malaria diagnosis (i.e., patients with high parasitemia) in hospitals in Africa. Although we believe that such devices may be useful to control malaria, the concept of device development is quite different from that of image cytometry in our manuscript. Consequently, to not mislead the readership, we think that we should not describe smartphone-based devices. 

Comment 3

Perhaps, the method would be of relevance for the detection of other parasites based on the same approach. The authors can provide some future directions and guide the researchers towards it.

Response

    Thank you for your important suggestion. We have expanded revised manuscript, accordingly (lines 239-244).   

Comment 4

The clinical evaluation trial of the method with an established predicate method used by healthcare would add much more value to the study. This could be a subject for future investigation by the authors. If the authors want, they can specify it in the text.

Response

    Thank you for your suggestion. We are now reconstructing the image cytometer for POCT in Africa, and it will take time to start the clinical evaluation trial of the image cytometry. Thus, we think that we should not specify it at the present.   

We would like to thank you and the referees for the helpful comments.

Round 2

Reviewer 1 Report

Thanks for working on revision. I think there is a confusion about standard methodology for determining sub-microscopic parasitemia. Please be very clear about this in the manuscript. A few minor grammatical errors, which if corrected can bring more clarity.

Author Response

I think there is confusion about the standard methodology for determining sub-microscopic parasitemia. Please be very clear about this in the manuscript.

Response

    Thank you very much for pointing out very important issues. Although various protocols for the determination of parasitemia in submicroscopic carriers have been reported, standardized protocols are lacking at present. In order to emphasize that Giemsa microscopy is the most appropriate reference method in this study, I think that these should be clearly described in our manuscript. Therefore, these were described in the revised manuscript (lines 196-200). 

Comment 2

A few minor grammatical errors, which if corrected can bring more clarity.

Response

    I asked a professional native speaker to proofread the manuscript.